# Extracellular-Ca^2+^-Induced Decrease in Small Molecule Electrotransfer Efficiency: Comparison between Microsecond and Nanosecond Electric Pulses

**DOI:** 10.3390/pharmaceutics12050422

**Published:** 2020-05-04

**Authors:** Diana Navickaite, Paulius Ruzgys, Vitalij Novickij, Milda Jakutaviciute, Martynas Maciulevicius, Ruta Sinceviciute, Saulius Satkauskas

**Affiliations:** 1Biophysical Research Group, Faculty of Natural Sciences, Vytautas Magnus University, Vileikos st. 8, LT 44404 Kaunas, Lithuania; diana.navickaite@vdu.lt (D.N.); paulius.ruzgys@vdu.lt (P.R.); milda.jakutaviciute@vdu.lt (M.J.); martynas.maciulevicius@vdu.lt (M.M.); ruta.sinceviciute@vdu.lt (R.S.); 2Faculty of Electronics, Vilnius Gediminas Technical University, 10221 Vilnius, Lithuania; vitalij.novickij@vgtu.lt

**Keywords:** calcium electroporation, calcium, microsecond electroporation, nanosecond electroporation, membrane repair, pore resealing

## Abstract

Electroporation—a transient electric-field-induced increase in cell membrane permeability—can be used to facilitate the delivery of anticancer drugs for antitumour electrochemotherapy. In recent years, Ca^2+^ electroporation has emerged as an alternative modality to electrochemotherapy. The antitumor effect of calcium electroporation is achieved as a result of the introduction of supraphysiological calcium doses. However, calcium is also known to play a key role in membrane resealing, potentially altering the pore dynamics and molecular delivery during electroporation. To elucidate the role of calcium for the electrotransfer of small charged molecule into cell we have performed experiments using nano- and micro-second electric pulses. The results demonstrate that extracellular calcium ions inhibit the electrotransfer of small charged molecules. Experiments revealed that this effect is related to an increased rate of membrane resealing. We also employed mathematical modelling methods in order to explain the differences between the CaCl_2_ effects after the application of nano- and micro-second duration electric pulses. Simulation showed that these differences occur due to the changes in transmembrane voltage generation in response to the increase in specific conductivity when CaCl_2_ concentration is increased.

## 1. Introduction

Electroporation is a physical method for exogenous molecule delivery through the plasma membrane that is used to increase the plasma membrane permeability by applying short (ns–ms) but strong electric pulses [1]. The permeability is increased due to the electric-field-induced transmembrane voltage, resulting in the formation of transient hydrophilic pores in the cell membrane [2]. Pore formation starts when transmembrane voltage exceeds an electroporation threshold ranging between 0.2 and 1 V [3,4]. During the prolonged application of the over-threshold electric field, the electropore density and/or size increases [5]. However, when the external electric field is switched off, the electric-field-induced transmembrane potential dissipates, initiating the process of pore resealing [6].

The presence of electropores—either reversible or irreversible—enables bidirectional transport across the cell membrane [7]. Such increased permeability can be used for the effective delivery of small charged molecules (e.g., fluorescent dyes [8] or membrane-impermeable chemotherapeutic drugs [9]) to the cells. The combined application of electroporation and chemotherapeutic agents is termed electrochemotherapy and is a successful method with clinically proven efficiency in treating different tumours [10]. In the recent years, intracellular calcium ion (Ca^2+^) delivery via electroporation has been suggested as an alternative to electrochemotherapy with bleomycin or cisplatin [11]. A recent clinical trial has shown that calcium electroporation has comparable efficiency as electrochemotherapy and is feasible and effective in patients with cutaneous metastases [12].

Ca^2+^ ions are universal signal mediators that regulate many cellular functions [13]. At physiological conditions, the concentration of Ca^2+^ ions in the cytoplasm ranges between 10^−8^–10^−7^ M and between 10^−3^–10^−2^ M in the extracellular environment. Calcium homeostasis is maintained by the interplay between multiple types of energy-dependent pumps and passive directional transporters located in plasma membranes and organelles [14]. Calcium ion distribution between extra- and intra-cellular compartments plays a crucial role in cellular response to various external stress conditions, and they are also involved in certain types of cell death, for instance, intracellular Ca^2+^ overload initiated necrotic and apoptotic processes [11,15].

Calcium is also known for its role in the resealing of an injured membrane [16]. There are several ways that Ca^2+^ ions are involved in membrane resealing, further delineating the importance of maintaining membrane integrity [17]. Therefore, in this study, we aimed to investigate the impact of different CaCl_2_ concentrations on the electrotransfer of small charged molecules. For this purpose, Chinese hamster ovary (CHO) cells were electroporated in media with different CaCl_2_ concentrations using either microsecond or nanosecond electroporation. Microsecond electroporation is the “classic” mode of electroporation and utilizes electric pulses of µs–ms duration. In contrast, the nanosecond duration pulses have their duration below 1 µs, and much higher pulse strengths. Due to the short duration, they can generate transmembrane potential on intracellular membranes instead or in addition to the plasma membrane [18,19] and are lucrative for medical applications because, unlike microsecond duration pulses, they do not induce muscle twitching [20].

Propidium iodide (PI), YO-PRO-1 and ethidium bromide (EtBr) were used as fluorescent probes to investigate the uptake of small charged molecules after electroporation. The obtained results show that CaCl_2_ presence in the electroporation media prior electric field application impedes the uptake of small charged fluorescent molecules through the plasma membrane.

## 2. Materials and Methods

### 2.1. Cell Line

Chinese Hamster Ovary (CHO) cells (European Collection of Authenticated Cell Cultures, 85050302) were used for in vitro experiments. CHO cells were cultivated in Dulbecco’s Modified Eagle Medium (DMEM) (Sigma, Darmnstadt, Germany) supplemented with 10% Foetal Bovine Serum (FBS) (Sigma), 1% l-glutamine (Sigma) and 1% penicillin-streptomycin solution (Sigma). The cells were grown in monolayers in 10 cm Petri dishes (Techno Plastic Products (TPP) (Trasadingen, Switzerland) and incubated at 37 °C in 5% CO_2_ atmosphere. To maintain the culture, cells were passed every 2–3 days and 24 h before the experiments.

### 2.2. Cell Electroporation

After harvesting, cells were washed and diluted in electroporation medium (10 mM 4-(2-hydroxyethyl)-1-piperazineethanesulfonic acid (HEPES) (Lonza, Basel, Switzerland), 250 mM sucrose (Sigma), 1 mM MgCl_2_ (Sigma) in sterile water) that was supplemented with different CaCl_2_ (Sigma) concentrations (0, 0.0001, 0.001, 0.01, 0.1, 0.25, 0.5, and 1 mM).

The HEPES buffer with 0 mM CaCl_2_ was used as the experimental control. Cell suspensions were prepared at a concentration of 1.8 × 10^6^ cells/mL, using buffers with different CaCl_2_ concentrations. For each experimental point, 6.5 × 10^4^ cells (36 μL) were used. Cuvette electrodes with a 1-mm gap between electrodes were used for cell electroporation. For small molecule electrotransfer experiments, the cell solution was supplemented with 4 μL propidium iodide (PI) (Sigma, Darmnstadt, Germany) at the final concentration of 40 µM, 4 µL ethidium bromide (EtBr) (Carl Roth, Karlsruhe, Germany) at the final concentration of 40 µM or 4 µL YO-PRO-1 (Thermo Fisher Scientific, Dublin, Ireland) at the final concentration of 4 µM. For the experiments to test the medium conductivity effect on the efficiency of PI electrotransfer CaCl_2_ was replaced by MgCl_2_.

For the experiments investigating pore resealing dynamics, PI was present in the electroporation medium during electroporation only for positive controls and was introduced into the solution at 15, 30, 60, 120, 360, 480 or 600 s after electroporation. As the PI can only enter the cells with compromised plasma membrane integrity, the percentage of PI positive cells in these experiments reflect the percentage of cells that did not reseal after electroporation. For micro-electroporation, CHO cells were electroporated using 1 square HV pulse (800–1800 V/cm pulse strength, 100 μs pulse duration) using BTX T820 electroporator (Harvard Apparatus, Holliston, MA, USA) For nano-electroporation, cells were electroporated using 10 square HV pulses (10,000–18,000 V/cm pulse strength, 200 ns pulse duration, 1 Hz repetition frequency) using high voltage pulse generator (VGTU, Vilnius, Lithuania). After 15 min incubation at 37 °C in 5% CO_2_ atmosphere, cell permeabilization was evaluated using flow cytometry (BD Accuri C6, BD Biosciences, Franklin Lakes, NJ, USA). A total of 10^4^ cells per sample at 66 µL/min flow rate and 22 µm core size were collected. The cells were excited with 488 nm laser and fluorescence was collected using 585/40- and 533/30-nm bandpass filters. The flow cytometry gating strategy to collect the cells after various conditions and for all three molecules used are depicted in Figure 1.

### 2.3. Visualization of Electroporation

For the visualization of PI electrotransfer experiments, 22 × 22 mm glass cover slips (Carl Roth, Karlsruhe, Germany) were immersed into 70% ethanol for 15 min. Then, the coverslips were removed from the ethanol and placed into the 4-cm Petri dishes and airdried for 15 min. Then, 40 µL of cell suspension (1.8 million per mL) was placed on the middle of the coverslip in the Petri dish. After an additional 5-min incubation (for the cells to settle down on the surface) and a gentle 2-mL addition of DMEM (supplemented with 10% FBS, 1% l-glutamine and 1% penicillin-streptomycin solution), the Petri dishes with cells were placed in the incubator (37 °C and 5% of CO_2_). After 5 h of incubation, the coverslips with cells attached (5 h incubation) was taken from the petri dishes and placed on the laboratory made electrodes (copper foil mounted on the objective glass) with a 2-mm gap. Electroporation medium (HEPES buffer supplemented with different concentrations of CaCl_2_ (0, 0.25, 0.5, 1 mM) and supplemented with PI (40 µM final concentration) was added between the gap of electrodes prior to the coverslip being mounted on the top of the electrodes. The cells were electroporated using 1 HV pulse (1400 V/cm or 1800 V/cm pulse strength, 100 μs pulse duration) delivered by a BTX T820 pulse generator. A Motic AE31 fluorescent microscope mounted with a MoticamPro 285B camera was used for cell imaging. For PI fluorescence imaging, a filter cube (D560/40X excitation, dichroic 595DCLP mirror, D630/60 emission) was used. Motic Images Advanced 3.2 software was used to obtain the images. At all conditions, phase contrast images were taken prior to the application of electric fields. For the fluorescence imaging time lapse was switched on before electroporation. Single HV pulse was delivered at time ‘0 s’. Then, additional fluorescent images were taken at every second for 300 s. Open source image processing software ImageJ (Version 1.52p, National Institute of Health, Bethesda, MD, USA) was used to calculate corrected total cell fluorescence (CTCF) [21]. The phase contrast image was used to determine the area of the cell, in which the changes of the propidium iodide fluorescence were measured over time. CTCF was calculated by multiplying the cell area (the area in phase contrast image delineated by single cell) with its average fluorescence. The changes in CTCF allowed to monitor the dynamics of PI entry. The CTCF mean values in a corresponding figure were made from the observation of at least 20 different cell images.

### 2.4. Model for Computation

The model is mathematically defined in Equations (1), (2) and (3) [22].
(1)$$\Delta \Phi_{m}\left(t\right)=fERcos\theta \;\left[1-e^{\left(-\frac{t}{\tau }\right)}\right]$$ where Δ*Φ* is transmembrane voltage, *f* is the shape factor, *R* is the cell radius, *θ* is the angle measured from the centre of the cell with respect to the direction of the field, *t* is the time elapsed since the onset of the field, and *τ* is the time constant of membrane charging.
(2)$$f=\frac{3\lambda_{ο}\left[3dR^{2}\lambda_{i}+\left(3d^{2}R-d^{3}\right)\left(\lambda_{m}-\lambda_{i}\right)\right]}{2R^{3}\left(\lambda_{m}+2\lambda_{ο}\right)\left(\lambda_{m}+\frac{1}{2}\lambda_{i}\right)-2{\left(R-d\right)}^{3}\left(\lambda_{ο}-\lambda_{m}\right)\left(\lambda_{i}-\lambda_{m}\right)}$$ where *λ_o_*, *λ_m_* and *λ_i_* are the conductivities of the external, membrane and cytoplasm, respectively, *R* is the radius of the cell and *d* is the thickness of the membrane.
(3)$$\tau =\frac{Rc_{m}}{\frac{2\lambda_{ο}\lambda_{i}}{2\lambda_{ο}+\lambda_{i}}+\frac{R}{d}\lambda_{m}}$$ where *τ* is the membrane charging time, *R* is the radius of the cell, *C_m_* is the capacitance of the membrane, *λ_o_*, *λ_m_* and *λ_i_* are the conductivities of the external, membrane and cytoplasm, respectively, and *d* is the thickness of the membrane.

In all the situations, the transmembrane potential change at cell poles (cos(0°) = 1) was calculated. The thickness of the membrane d was 3 × 10^−9^ m, as described in [23], the cytoplasm conductivity *λ_i_* was 0.5 S/m, as described in [24], membrane conductivity *λ_m_* was 3 × 10^−7^ S/m as described in [22] and membrane capacitance *C_m_* was 10^−2^·F·m^−2^ as described in [22]. The average cell radius *R* was experimentally evaluated by calculating the radii of >100 cells in microscopy pictures using open-source image processing program ImageJ. The *R* value determined this way was 9.7 µm. The specific conductivity values of the extracellular media with different CaCl_2_ concentrations were measured and the results are presented in Table 1.

### 2.5. MTT Assay

3-(4,5-Dimethylthiazol-2-yl)-2,5-Diphenyltetrazolium Bromide (MTT) (Carl Roth, Karlsruhe, Germany) assay was performed to evaluate long term cell response after cell electroporation in the presence of CaCl_2_. MTT is used as a colorimetric cell viability assay, which relates the enzymatic activity of the cell with its viability. The colorimetric assay is based on ability of NAD(P)H-dependent cellular oxidoreductase enzymes to reduce the yellow tetrazolium dye MTT to its insoluble purple formazan. After cell treatment with electric pulses in electroporation medium, supplemented with different CaCl_2_ concentrations (0, 0.25, 0.5, 1 mM), 9000 cells in 200 µL of growing media were plated in each well of 96-well microplates (Plastibrand; Wertheim, Germany) and incubated at 37 °C in 5% CO_2_ atmosphere. After 24 h of incubation, 20 µL of growing medium was removed from the wells and 20 µL of MTT salt at concentration of 0.5 mg/mL was added and incubated for additional 2 h. Afterwards, the medium was taken out from the wells and the wells were washed twice with 100 µL of Phosphate-Buffered Saline (PBS) (Carl Roth, Karlsruhe, Germany) Formazan formed in the cells was dissolved by using 100 µL of DMSO (Sigma). A multiwell scanning spectrophotometer (spectro star nano BMG Labtech, Ortenberg, Germany) was used to measure the absorbance of the samples in microplate. Optical density was estimated at 570 nm. All experimental points were normalised to the control (untreated cells).

### 2.6. Flow Cytometry Assay (FCA)

The flow cytometer (BD Accuri C6, BD Biosciences, Franklin Lakes, NJ, USA) modality to determine the number of cells in a specific volume was employed in the study to estimate exact number of cells in a specific sample at 15 min time points after the cell treatment with electric pulses in electroporation medium, supplemented with different CaCl_2_ concentrations. A flow cytometer value of obtained cell speed was obtained when 10,000 cells from the sample were measured. A mean of measured cells in control samples was normalised to a 100%. Other samples were normalised according to the control (untreated cells).

### 2.7. Statistical Analysis

Experiments for each individual experimental point were repeated 3 times on at least two separate days. The results are presented as mean ± standard mean error (SEM). For visualization, 20 cells per field of view were used to calculate the average CTCF. To test significance, one-way ANOVA with a Bonferroni post-hoc test was used for all experiments. The assumption for normality distribution was verified with Shapiro–Wilk normality test by setting the p value for rejection to 0.05. The statistical analysis was performed using Sigma Plot 12.5 software. The percentages were transformed using logit transformation before the ANOVA analysis.

## 3. Results

The first set of experiments was designed to evaluate whether the addition of CaCl_2_ to the electroporation medium can have any influence on the PI electrotransfer efficiency after microsecond range electroporation. The dependence of PI positive cell percentage on the CaCl_2_ concentration in the electroporation medium is summarized in Figure 2A, and the total fluorescence of PI positive cells is shown in Figure 2B. These results show that CaCl_2_ reduce the efficiency of PI electrotransfer. Even the lowest CaCl_2_ concentration used (0.25 mM) has significantly reduced both the amount of PI positive cells and their total fluorescence. For example, using 1600 V/cm electric pulse strength in a medium without added calcium, PI positive cell percentage is around 69%. However, 1600 V/cm electric pulse in the electroporation medium with 0.25 mM CaCl_2_ yields only ~28% of PI positive cells. The increase in CaCl_2_ concentration from 0.25 to 0.5 and 1 mM did not further decrease the amount of PI positive cells or their total fluorescence. The total cell fluorescence depicted in Figure 2B shows similar trends. Irrespectively of the electric pulse strength, the total fluorescence of PI positive cells that were electroporated with CaCl_2_ (0.25, 0.5, 1 mM) is ~2 times lower than that of the cells electroporated in the control medium (0 mM CaCl_2_). It should also be noted that, after increasing the electric pulse strength from 1600 to 1800 V/cm, the percentage of PI positive cells increases more sharply than the total fluorescence of the cells.

Our results indicate that PI entry into the cells electroporated with microsecond electroporation parameters is significantly diminished when CaCl_2_ is added to the electroporation medium prior to the application of electric field. The next set of experiments was performed in order to determine whether the same effect is visible when nanosecond electroporation was used. The obtained results are presented in Figure 3 (A—percentage of PI positive cells, B—total fluorescence of PI positive cells).

The effect of the CaCl_2_ concentration on the PI electrotransfer efficiency after nanosecond electroporation can be most clearly seen at 14,000–15,000 V/cm of electric field strength. At these conditions, there are no significant differences in PI positive cell percentage when the cells were electroporated in the media with 0.25 and 0.5 mM CaCl_2_ concentrations. Cells electroporated in either of these media resulted in significantly lower amount of PI positive cells (~25%) when comparing to the cells electroporated in control medium (0 mM CaCl_2_, ~77%). A further increase in CaCl_2_ concentration to 1 mM leads to the percentage of PI positive cells (~45%) that is significantly increased from the one observed with 0.25 and 0.5 mM CaCl_2_ concentrations, but significantly lower than the one observed in the control (0 mM CaCl_2_). Therefore, it can be summarized that CaCl_2_ reduces the efficiency of PI electrotransfer using both microsecond (Figure 2) and nanosecond (Figure 3) pulses. However, using nanosecond electric pulses, this reduction is lower when a higher concentration of calcium is used, showing the likelihood of a secondary effect of CaCl_2_ on nanosecond electroporation.

Interestingly, the effect of CaCl_2_ is the same with all CaCl_2_ concentrations used for microsecond duration pulses and the same with 0.25- and 0.5-mM CaCl_2_ concentrations for nanosecond duration pulses. This shows that the calcium induced reduction in PI electrotransfer efficiency might be based on external CaCl_2_ concentration threshold that is below 0.25 mM. Alternatively, this effect might be dependent on the concentration but has already reached a plateau at 0.25 mM. To elucidate this, we electroporated the cells with 1 × 1400 V/cm strength and 100 µs duration electric pulse or 10 × 1.4 kV/cm strength and 200 ns electric pulses repeated at 1 Hz frequency with an expanded range of CaCl_2_ concentrations (0.0001–1.0 mM). The results of these experiments are presented in Figure 4. We see a gradual decline in the number of PI positive cells (Figure 4A) and total fluorescence (Figure 4B) with microsecond-duration electric pulses as they were well approximated by the exponential function (*R*^2^ = 0.95 and *R*^2^ = 0.86, respectively). However, the approximations failed for the results of PI electrotransfer that were obtained using nanosecond-duration electric pulses (Figure 4C,D). This indicates gradual continuous PI positive cell dependence on CaCl_2_ concentration for microsecond electroporation and a concentration threshold effect for nanosecond electroporation. The threshold for nanosecond electroporation was determined to be between 0.1 and 0.25 mM CaCl_2_ concentration, as indicated by the results of PI positive cells (Figure 4C) and total cell fluorescence (Figure 4D).

The next step in our investigations was to assess if CaCl_2_ induced reduction in electrotransfer efficiency is caused by a specific feature of propidium iodide or is it applicable to electrotransfer of other small charged molecules as well. To assess this, we electroporated the cells with 1 × 1400 V/cm strength and a 100-µs duration electric pulse or 10 × 1.4 kV/cm strength and 200-ns electric pulses repeated at 1 Hz of frequency electric pulses in media containing 0, 0.25 or 1 mM CaCl_2_ concentration in the presence of three different small charged molecules: propidium iodide (molecular weight 668.36 g/mol, formal charge +2), YO-PRO-1 (molecular weight 629.32, formal charge +2) and ethidium bromide (EtBr, molecular weight 394.29 g/mol, formal charge +1). The results of these experiments are presented in Figure 5A (microsecond pulses) and Figure 5B (nanosecond pulses). It can be seen that, after microsecond duration electric pulses, the electrotransfer efficiency (Figure 5A) decreases for all three molecules tested. The highest drop in both the electrotransfer efficiency is observed with EtBr. However, a different situation is observed with nanosecond electric pulses (Figure 5B). It can be seen that while the decrease in electrotransfer efficiency (Figure 5B) is observed with YO-PRO-1 and PI, the electrotransfer efficiency increases when EtBr is used.

In order to obtain a deeper understanding of the phenomenon of the CaCl_2_-mediated inhibition of PI electrotransfer, we visualized the dynamics of PI electrotransfer into the cells after microsecond pulse treatment using electroporation media with different CaCl_2_ concentrations (0, 0.25, 0.5, 1 mM). The results of PI electrotransfer dynamics after electroporation are shown in Figure 6.

These results show that PI electrotransfer into the cells with CaCl_2_ present in electroporation media is disturbed from the first seconds after electroporation when comparing to cells without CaCl_2_ using both 1400 (Figure 6A,C) and 1800 V/cm (Figure 6B,D) electric field strengths. It can be seen that PI electrotransfer is significantly lower with all electric parameters and all CaCl_2_ concentrations tested in comparison to PI electrotransfer to cells in media without CaCl_2_. However, using a 1800-V/cm pulse strength, the results show that the medium with 0.25 mM CaCl_2_ yields a significantly higher PI entry in comparison to media with 0.5 or 1 mM CaCl_2_. The representative pictures of cells electroporated at 1400 (Figure 6C) and 1800 V/cm (Figure 6D) in the media with different CaCl_2_ concentrations visually illustrate the data presented in Figure 6A,B, providing visual proof that the fluorescence of the cells decreases with increasing CaCl_2_ concentration.

One of the possible ways in which the calcium ions could exert its effect on the electroporation is the increased rate of membrane resealing, leading to diminished molecular transport across the membrane. To test this theory, we decided to investigate the dependence of pore resealing dynamics on the presence of extracellular CaCl_2_ concentration by monitoring the percentage of PI positive cells when PI is added after the application of electric pulses. The results of these experiments are presented in Figure 7. A single HV pulse with 1400 V/cm (Figure 7A) or 2800 V/cm (Figure 7B) strength and 100 µs duration was used. The results after electroporation with 1400 V/cm strength electric pulses showed that PI transfer to the cells is reduced immediately after application of electric fields in the presence of extracellular Ca^2+^ ions. Therefore, it can be assumed that these conditions either cause immediate membrane resealing (pore closure and/or membrane repair) or impede the pore formation. However, this is not the case when electric pulses with double (2800 V/cm) electric field strength are used. In these conditions, a clear dependence of the duration of the membrane resealing on the concentration of extracellular Ca^2+^ is observed. Indeed, it can be seen that 0.25 mM of extracellular CaCl_2_ was enough to significantly diminish the membrane resealing duration.

However, it is interesting to note that while we see the calcium-induced decrease in PI electrotransfer, there is a significant difference in the trend of PI electrotransfer between the cells electroporated with micro- and nano-second duration pulses. After a certain threshold in CaCl_2_ concentration, the PI electrotransfer efficiency increases again using nanosecond duration electric pulses, while no such effect is observed using microsecond duration pulses (Figure 2 and Figure 3). For a better understanding of this inversion, we have utilized the widely accepted model of transmembrane voltage induction by electric field [22]. The model neglects possible cell deformation during prolonged electric field application [24,25].

The results of the modelling (Figure 8) show that the specific conductivity of the extracellular medium does not significantly change the generated transmembrane potential when microsecond duration pulses were used. However, a dramatic transmembrane potential change is observed when the simulations were performed using nanosecond duration pulses. The modelling results showing the transmembrane potential distribution on the whole surface of the cell are presented in Figure 8A and the peak transmembrane potential at the electrode-facing poles (cos*θ* = 0) is presented in Figure 8B (microsecond duration pulses) and Figure 8C (nanosecond duration pulses). It is clearly observed that, using nanosecond duration pulses, the peak of transmembrane potential increases approximately twice in all parameters used when extracellular the CaCl_2_ concentration changes from 0 to 1 mM.

Moreover, one can suggest that the observed effects are related with changes in medium conductivity. To test this possibility, we performed corresponding experiments with medium conductivities ranging from 0.016–0.16 S/m, where CaCl_2_ in the medium was replaced by MgCl_2_. The results showed that medium conductivity, in this range, has no effect on PI electrotransfer efficiency Figure 9.

It is also possible that Ca^2+^ can induce cell death in a very short time range after cell electroporation, and the observed decrease in PI positive cells could be because of the disintegration of those cells. However, cell count by flow cytometry assay (FCA), as described in [26], shows no cell disintegration in the range of CaCl_2_ concentrations used (Figure 10). In addition, MTT assay performed 24 h after cell treatment with 1 HV pulse (1400 V/cm, 100 µs) demonstrates no additional cell death at 0.25 and 0.5 mM compared to that at 0 mM CaCl_2_ (Figure 10).

## 4. Discussion

In the current research, CHO cells were electro-permeabilized in the presence of different extracellular CaCl_2_ concentration in order to investigate the effect of Ca^2+^ on the electrotransfer efficiency of small charged molecules. The presented results clearly demonstrate that the electrotransfer efficiency of PI is impeded when calcium is present in electroporation medium. This process is dependent on the concentration of calcium ions, although a plateau is apparent at relatively low calcium concentrations. Notably, the phenomenon is observed after electroporation with both microsecond duration and nanosecond duration pulses with some differences.

To eliminate possibility that this effect is not specific to PI only, for example, due to a calcium-mediated decrease in PI fluorescence or calcium-mediated inhibition of PI and DNA binding, we performed similar experiment with two other small hydrophilic molecules, namely EtBr and YO-PRO-1. After microsecond electroporation, the reduction in the electrotransfer efficiency is observed with all three different molecules—PI, EtBr and YO-PRO-1—investigated in this study. However, after nanosecond electroporation, both the electrotransfer efficiency and the total fluorescence of the cells markedly decreased for PI and YO-PRO-1 but increased for EtBr. This is likely related to the size and the charge of the molecules. PI and YO-PRO-1 both have similar molecular weights and +2 formal charge. In contrast, EtBr has ~1/3 lower molecular weight and +1 formal charge, which likely changed its behaviour after nanosecond electroporation.

In order to understand the observed differences, we performed the simulation of transmembrane voltage generated upon delivery of microsecond and nanosecond duration pulses. Simulation showed that with an increase in CaCl_2_ concentration, a higher transmembrane voltage was induced when nanosecond, but not microsecond, pulses were used. Most probably, this is related with the increase in electroporation medium conductivity (see Table 1) and consequently the decrease in membrane charging time. Indeed, since the cell membrane charging time is below 5 µs [24], the delivery of ten 200-ns pulses at 1 Hz of frequency might result in incomplete membrane charging. Therefore, although calcium ions decrease the efficiency of small molecule electrotransfer, this decrease can be compensated by increase in transmembrane voltage generated upon the delivery of nanosecond pulses. According to simulation, increases in CaCl_2_ concentration do not have any effect on the generation of transmembrane voltage when microsecond pulses are used. Therefore, the observed effect with microsecond pulses is attributed solely to calcium ions.

Calcium ions play multiply roles in cell physiology; however, when analysing the role of CaCl_2_ in the decrease in efficiency of small charged molecules, electrotransfer pore formation and annihilation must be considered. The cell after electroporation faces a critical threat to its viability in the form of uncontrolled transport of essential ions and organic molecules through the electric field affected membrane. Therefore, defense mechanisms that rapidly repair plasma membrane lesions have to be employed in order to maintain cell viability after membrane permeabilization. The plasma membrane repair requires membrane replacements, fusion events and cytoskeleton reorganization [27]. It can be segmented into passive and active membrane repair, both of which are triggered by Ca^2+^ ion influx at the injury site due to a thousand-fold gradient of calcium that exists across the plasma membrane [28].

It has been shown that Ca^2+^ ions interact with phospholipid heads, leading to phospholipid bridging via hydrophobic bonding [29]. The phospholipid bridging initiates the aggregation of the phospholipids, which provides a significantly higher chance for the fusion of the vesicles, thus allowing phospholipids in damaged cell membrane areas to fuse. This way, there is a greater chance for a damaged cell membrane to reseal.

Another Ca^2+^-ion-induced phenomenon that was observed in previously published experiments with liposomes is the reduction in the fluidity of the membrane induced by Ca^2+^ interaction with the phospholipid heads [30,31]. The membrane fluidity term describes a relative diffusion motion of molecules within membranes [32]. Some studies had been investigating the relationship between the membrane fluidity and the electroporation [33,34]. In these, the membrane fluidity was changed by regulating the cholesterol percentage in the membrane [35,36]. The studies have indicated that the electroporation threshold is dependent on the membrane fluidity [33,34]. Indeed, the transmembrane potential threshold for electroporation has an inverse correlation with the fluidity of the lipid bilayer [37]. Therefore, if calcium ions are present in the extracellular medium, then the fluidity of the membrane is decreased, increasing the transmembrane potential threshold required for electroporation in turn. This assumption also goes in agreement with the published simulations addressing the dependence of electropore formation on Ca^2+^ ions [38]. Here, we show that higher applied electric pulse strength is needed in order to obtain the similar amount of PI positive cells in Ca^2+^-containing medium in comparison to the medium without added Ca^2+^, which is consistent with this hypothesis.

Moreover, it has also been demonstrated that phospholipid aggregates induced by Ca^2+^ ion bridging increase the phospholipid aggregate repulsiveness to water molecules [39]. The change in phospholipid–water interactions becomes relevant when using hydrophilic pore formation model to explain electroporation. According to this model, pores do not immediately close after the electric-field-induced transmembrane potential dissipates [22]. Instead, after the electric field is switched off, the size of the pores rapidly diminishes to ~0.4–2.3 nm [22,40]. Taking the reduced pore size and the model of the phospholipid head lined interior of the pore into the account, the repulsion to water by the Ca^2+^ ion bridging of phospholipid aggregates can make a great impact on the transfer of hydrophilic molecules.

Additionally, extracellular calcium, intracellular vesicles and calcium-dependant exocytosis are involved in the active membrane repair [41]. Specifically, calcium and cytoplasmic vesicles have been identified as a part of repair/resealing machinery [21]. This repair machinery is initiated via annexins, a family of Ca^2+^ regulated proteins [42,43]. However, the exact mechanism of the annexin role in the repair of membrane damage is not yet clearly understood. Nevertheless, it has been shown that annexins 1 and 2 interact with dysferlin, a protein involved in membrane resealing after dramatic damage [44,45]. The calcium-triggered assembly of 2D arrays of annexin 5 has been also observed to play a key role in cell membrane repair [46].

One could also suggest that Ca^2+^ influx can close non-specific ion channels that permit dye uptake. Indeed, it has been shown that Ca^2+^ overload in the cell desensitizes many cation channels [47,48,49]. Nevertheless, these channels do not permit the transmembrane diffusion of dyes like propidium iodide. On the other hand, connexin hemichannels allow for the exchange of small molecules between the cytoplasm and the extracellular space [50]. This opens the possibility that Ca^2+^ can regulate PI uptake through these hemichannels. Nevertheless, studies on gating of various connexin hemichannels show that these hemichannels open under low extracellular calcium concentrations [51,52], i.e., in conditions that are opposite to the ones described above.

## 5. Conclusions

In conclusion, we report that extracellular calcium induces a negative effect on the small charged molecule electrotransfer into the cells due to electroporation. This effect was demonstrated by using conventional microsecond duration pulse electroporation. All used concentrations of extracellular CaCl_2_ had the negative effect small charged molecule electrotransfer. However, the effect was diminished when CaCl_2_ concentration increased from 0.25 to 1 mM using nanosecond duration pulse electroporation. This can be explained by mathematical modelling, which proves that, with nanosecond electric pulses, the increase in specific conductivity due to the higher concentration of CaCl_2_ results in significantly higher transmembrane potential generated on the cell. However, no change in the transmembrane potential generated on cells in media with different CaCl_2_ concentrations is observed when microsecond electric pulses are used. These results underline the differences between micro- and nano-second pulses used for electrotransfer of small charged molecules in the presence of CaCl_2_.

## Figures and Tables

**Figure 1 pharmaceutics-12-00422-f001:**
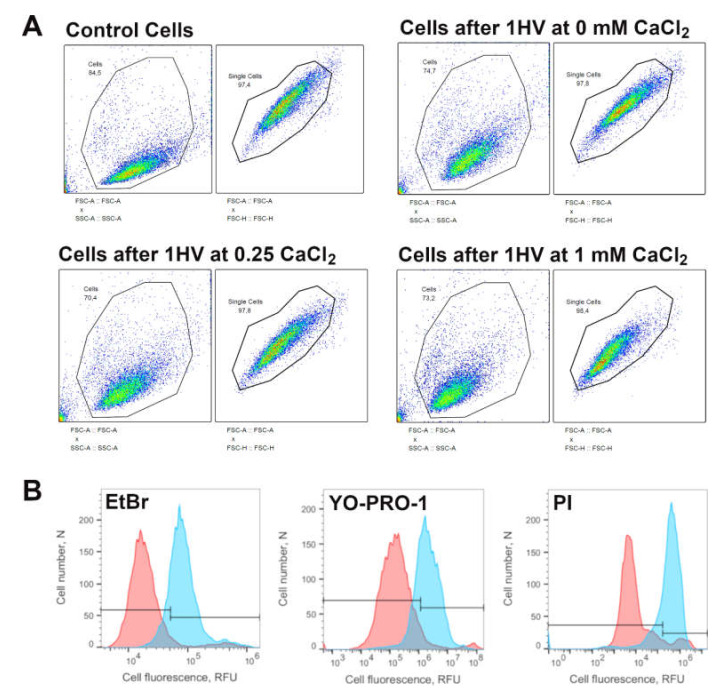
Flow cytometry gating strategies. Panel (**A**) represents cell distinction from the debris (FSC-A::SSC-A) and single cell (FSC-A::FSC-H) gating strategies. Panel (**B**) represents a gating strategy of cell fluorescence before (red) and 15 min after (blue) cell electroporation with HV pulse (1400 V/cm, 100 µs) in the presence of EtBr, YO-PRO-1 and PI molecules in electroporation medium containing 1 mM of CaCl_2_.

**Figure 2 pharmaceutics-12-00422-f002:**
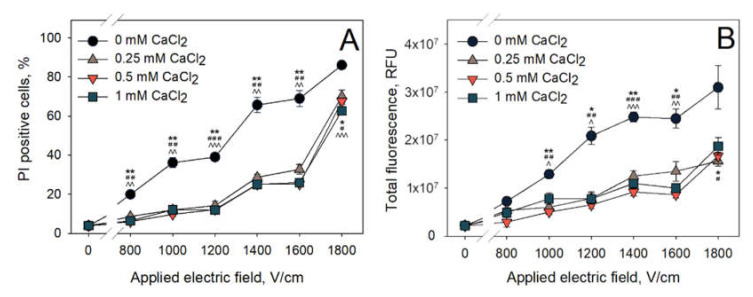
Dependence of PI electrotransfer efficiency (**A**) and total cell fluorescence (**B**) of the treated cells on the applied electric field strength at various CaCl_2_ concentrations. Cells were treated using 1 square HV pulse at a 100-μs pulse duration. PI fluorescence was measured 15 min after electric field application. The statistical differences between PI uptake (PI positive cells and total fluorescence) between control (0 mM CaCl_2_ concentration) and 0.25, 0.5, 1 mM CaCl_2_ concentrations are denoted by *, # and ^, respectively. One symbol denotes *p* < 0.05, two symbols—*p* < 0.01, three symbols—*p* < 0.001. The error bars represent the mean ± standard error of mean of *n* = 6 experimental replicates.

**Figure 3 pharmaceutics-12-00422-f003:**
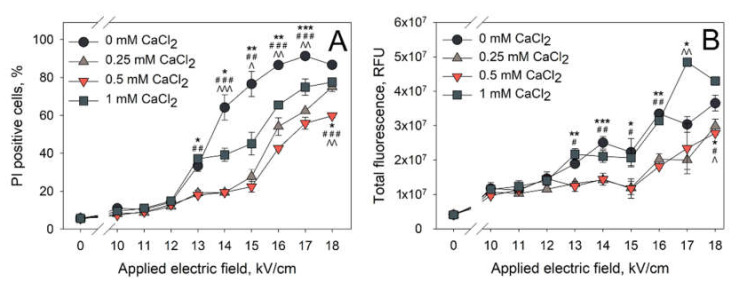
Dependence of PI electrotransfer efficiency (**A**) and total cell fluorescence (**B**) of the treated cells on the applied electric field strength at various CaCl_2_ concentrations. Cells were treated using 10 square HV pulses at a 200-ns pulse duration. PI fluorescence was measured 15 min after electric field application. The statistical differences between PI uptake (PI positive cells and total fluorescence) between control (0 mM CaCl_2_ concentration) and 0.25, 0.5, 1 mM CaCl_2_ concentrations are denoted by *, # and ^, respectively. One symbol denotes *p* < 0.05, two symbols—*p* < 0.01, three symbols—*p* < 0.001. The error bars represent the mean ± standard error of mean of *n* = 6 experimental replicates.

**Figure 4 pharmaceutics-12-00422-f004:**
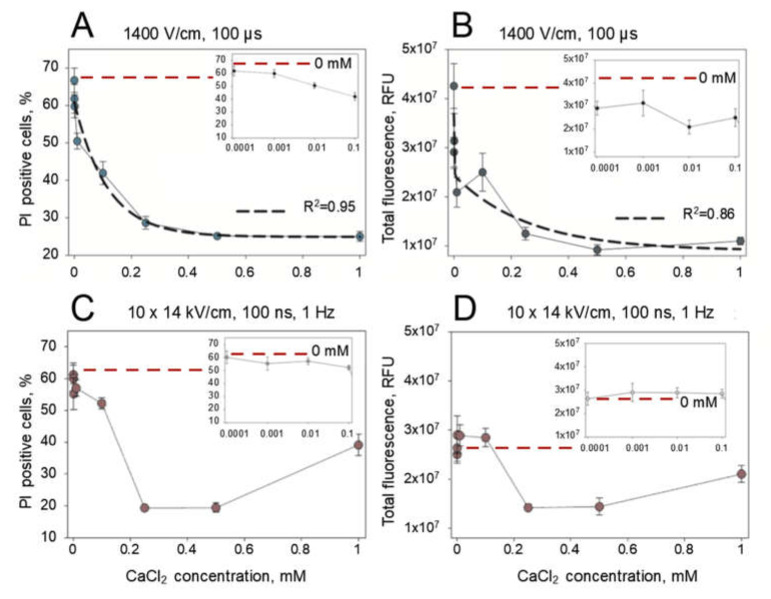
Dependence of PI electrotransfer efficiency (**A,B**) and the total cell fluorescence (**C,D**) after treatment with 1 × 1400 V/cm strength and a 100-µs duration electric pulse (**A,C**) or 10 × 1.4 kV/cm strength and a 200-ns electric pulses at 1 Hz of repetition frequency (**B,D**) at various CaCl_2_ concentrations; the results are represented in linear scale. The inserts in each figure represent the corresponding results in the range of low 0.0001–0.1 mM CaCl_2_ concentrations on the logarithmic scale. PI fluorescence was measured 15 min after electric field application. The error bars represent the mean ± standard error of mean of *n* = 6 experimental replicates.

**Figure 5 pharmaceutics-12-00422-f005:**
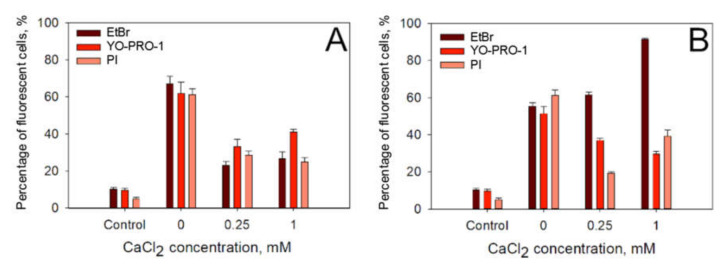
Dependence of PI, EtBr and YO-PRO-1 electrotransfer efficiency on the presence of extracellular calcium after treatment with 1 × 1400 V/cm strength and a 100-µs duration electric pulse (**A**) on the presence of extracellular calcium after treatment with 10 × 14 kV/cm strength and 200-ns duration electric pulses at 1 Hz of repetition frequency (**B**). Fluorescence was measured 15 min after the applied electric field. The error bars represent the mean ± standard error of mean of *n* = 6 experimental replicates.

**Figure 6 pharmaceutics-12-00422-f006:**
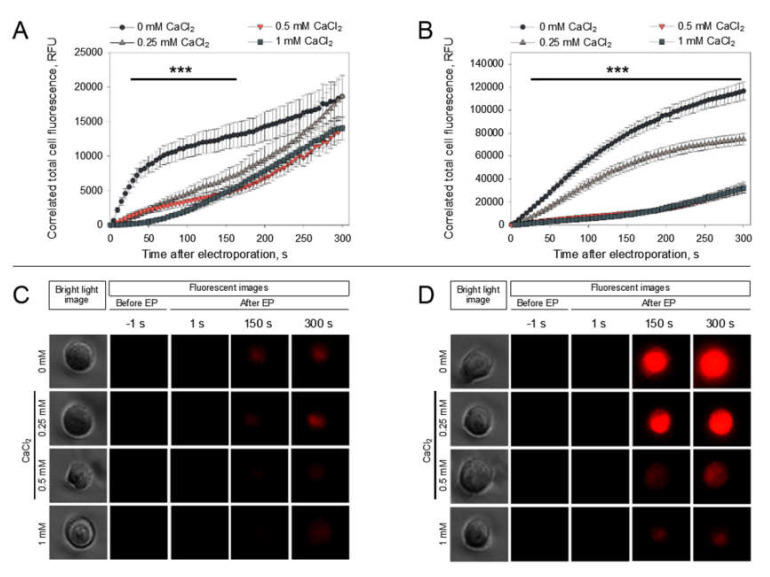
Visualization of PI electrotransfer into cells in media with different CaCl_2_ concentrations. The cells were electroporated with single 1400 V/cm (**A**) or 1800 V/cm (**B**) strength, 100 µs duration electric pulses. Electroporation was performed at time ‘0 s’. The fluorescent images in panels (**C**) and (**D**) represent key points images from the (**A**) and (**B**) panels, respectively. Statistical differences (A and B) of PI uptake between control (0 mM CaCl_2_) and 0.25; 0.5; 1 mM CaCl_2_ are denoted by ***—*p* < 0.001. The error represents the mean ± standard error of mean of *n* = 20 experimental replicates.

**Figure 7 pharmaceutics-12-00422-f007:**
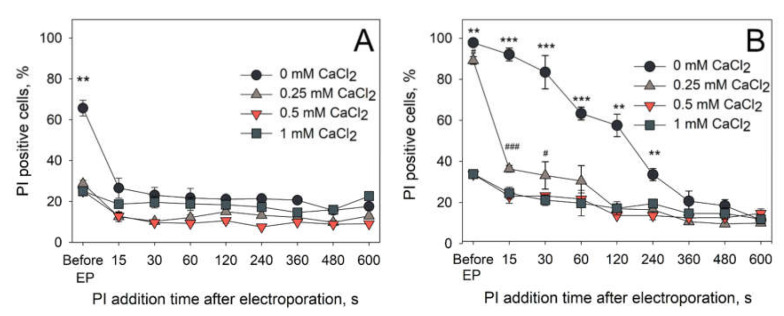
Membrane resealing dynamics after the treatment with a single 1400- (**A**) or 2800-V/cm (**B**) strength, 100-µs duration HV pulse, monitored by the entry of PI (40 µM) added 15–600 s after electroporation. Statistical differences between the control (0 mM CaCl_2_ concentration) and either of the 0.25, 0.5 or 1 mM CaCl_2_ groups are denoted as *, and between 0.25 mM CaCl_2_ and either 0.5 or 1 mM CaCl_2_ groups are denoted as #. A single symbol denotes two-tailed *p* < 0.05, double symbol—*p* < 0.01, and triple symbol—*p* < 0.001, PI fluorescence was measured 15 min after electric field application. The error bars represent the mean ± standard error of mean of *n* = 6 experimental replicates.

**Figure 8 pharmaceutics-12-00422-f008:**
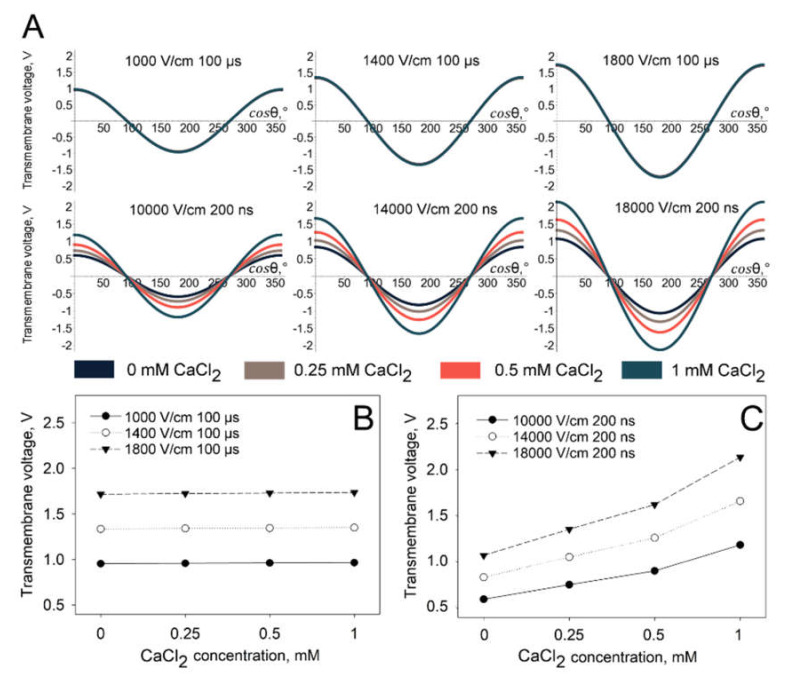
Modelling of transmembrane potential distribution on cell surface (**A**) and the peak transmembrane potential at the electrode-facing poles after microsecond (**B**) and nanosecond (**C**) electric pulse treatment.

**Figure 9 pharmaceutics-12-00422-f009:**
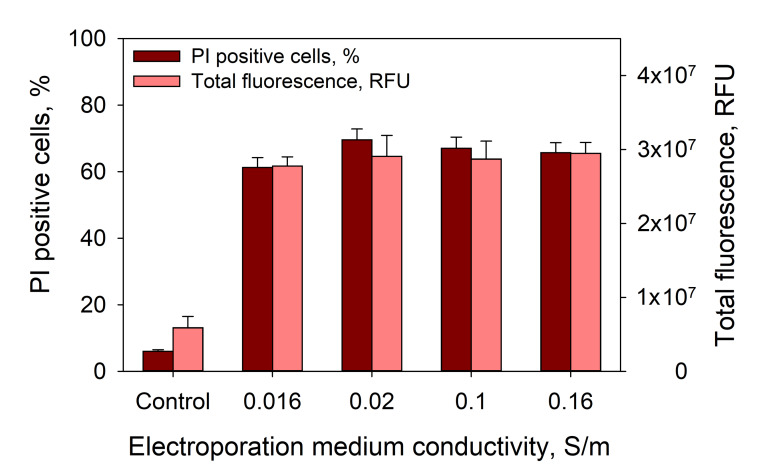
Dependence of PI electrotransfer efficiency and total cell fluorescence of the treated cells on the medium conductivity. Cells were treated with 1 × 1400 V/cm strength and a 100-µs duration electric pulse. The conductivity of the CaCl_2_-free medium was adjusted by adding MgCl_2_. The control represents untreated cells. PI fluorescence was measured 15 min after the applied electric field. The error bars represent the mean ± standard error of mean of *n* = 6 experimental replicates.

**Figure 10 pharmaceutics-12-00422-f010:**
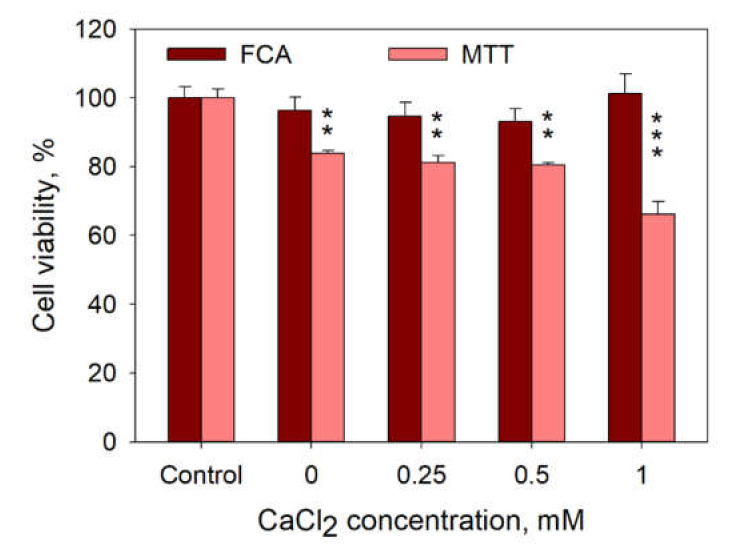
Cell viability after cell treatment using 1 HV pulse at a 100-μs pulse duration evaluated by flow cytometry assay (FCA, dark bars) and MTT (light bars) in dependence of the CaCl_2_ concentration in the medium. Symbols denote statistical differences between the control (untreated cells) and electroporation groups at 0, 0.25, 0.5, 1 mM CaCl_2_ concentration. Two symbols denote *p* < 0.01, three symbols—*p* < 0.001. The error bars represent the mean ± standard error of mean of *n* = 6 experimental replicates.

**Table 1 pharmaceutics-12-00422-t001:** The measured specific conductivity of used electroporation media containing different concentrations of CaCl_2_. Measurements were done using conductometer (Mettler Toledo S230).

CaCl_2_ Concentration in the Medium	Specific Conductivity
0 mM	0.016 S/m
0.25 mM	0.021 S/m
0.5 mM	0.026 S/m
1 mM	0.035 S/m

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
