# Peer review of "Extracellular-Ca2+-Induced Decrease in Small Molecule Electrotransfer Efficiency: Comparison between Microsecond and Nanosecond Electric Pulses"

_pharmaceutics, 2020, doi:10.3390/pharmaceutics12050422_

Round 1

Reviewer 1 Report

Authors report on an in vitro characterization study on the electrotransfer efficiency of small dye molecules under micro and nanosecond electric pulses in presence of extracellular calcium concentration.

The rationale has been clearly stated, the experimental procedures are technically sound and well described. The results have been adequately presented and discussed. I have only some suggestions:

  • I don’t see in the manuscript a direct connection between the experiments carried out and the phenomenon of calcium electroporation: in this paper, authors electro-permeabilize CHO cells to study the effects of different extracellular calcium concentrations on the electrotransfer of dye molecules; in the calcium electroporation, calcium is delivered into the cells via electropermeabilization. Authors must discuss this more clearly in the manuscript.

  • I suggest to rephrase the abstract to make more clear the aim and the results of the study.

  • I don’t understand the rationale for including results of Fig.9 and Fig.10 under the “Discussion” paragraph. This is also unusual.

Minor points:

  • Page 4, line 149: replace “a mean of measured in control samples” with “a mean of measured cells in control samples”;
  • Page 6, line 220: replace “100250” with “100-250”;
  • Page 8, line 260: replace “Figure 3” with 2Figure 6”;
  • Page 12, in Figure 9 caption: replace “Cells were treated using after treatment with….” with “Cells were treated with….”;
  • Page 14, line 454: delete “µs vs ns comparison”..

Reviewer 2 Report

The paper presents the effect of CaCl2 coupled to fluorescent dyes on cells in suspension in two electroporation conditions.

The paper has to be reorganized since some methods are described in results and discussion sections instead of in Material and Method section. Please, describe accurately all the experiments in the Material and Method section.

suggstions:

-introduction: line 36 anfd line 41 please rephrase 

- Material and method

  • Please, add the description of the experiments reported in results and discussion section as well as the description of the numerical mthod used in computation
  • linne 98 what is the interval 15-600 s? is a time t with t in the rane 15-600 s?
  • line 106 what represent fig. 1 ? they are results? why do you report them here? Please, discuss the meaning ogf the figure
  • Why three different fluorescent dye ? Please, discuss this topic.
  • line 118: please clirify the sentence 'between laboratory made....'
  • line 124: what type of images do you acquire at microscope: brigth field or phase contrast?
  • line 126 Do you acquier 300 images? How do you process them?
  • line 129: what is the meaning of 'the area of the cell' in your contest?
  • line 143: Please clirify the sentence ' Changes in optical density....'
  • line 147: why ist is important the exact number of cells?
  • line 152-157: do you describestatistical methods: for which analysis were used? the same in all the cases?

-Results

  • line 159: please, add the the experiment hypothesis before the results description
  • line 162-164 :Please clirify these lines ' these results show that..... While ....'
  • line 217: authors tell abut threshold: what kind of threshold? concentration? electric field?
  • line 220: what is 100250 uM?dose threshold?
  • line 257: How the described mechanism is evaluated?
  • line 260: I think do you refer to Fig. 6
  • Line 281-285: here authors describe methods. Plese move to specific paragraph
  • line 284: authors use 2800V/cm. they are sure that is not irreversible electroporation?
  • Why authors use only one pulse in electroporation?
  • line 302-328: authors describe a model for computation. Please, move in Material and Method section
  • line 324: please provide typical values for Cm
  • line 332-335. Authors refer to simulations or experiments? simulations were validated experimentally? please describe
  • from line 341: this part is discussion. please, move in the correct section. Please, refer to the concept that the current density is proportional to the local electric field and to the local conductivity.
  • line 345: the impedance was introduced: where it is measured? impedance is an integral quantity

- Discussion

  • Line 372: Please cliry the entence '....in order to maintain the viability of the electric field affected '
  • line 384: Why do you chosen MgCl2? please, describe the experiment in Material and Method section. Whit what electroporation conditions?
  • Line 392-404 Please, clarify do you introduce in your discussion the membrane fluidity
  • lines 422-429 Why it is of interest PI regulation with CaCl2. And the other fluorescent dyes why they were investigated?
  • fig. 10: please define FCA acronym
  • line 442: the line after could be entitled 'conclusion'

Minor suggestion:

line 106: 'to collect the repeated two times

line 336: 'of the cells IS...', ARE according to subject

Please, revise English language since in some points is difficult to read

Reviewer 3 Report

The manuscript by Navickaite et al shows some interesting results about the effects of calcium on permeabilisation of the plasma membrane after electroporation using microsecond and nanosecond pulses. However, several changes are needed.

General comments:

The written English needs improvement

Introduction:

l. 50-51: It has not been shown that the delivery of calcium has selectivity towards tumor cells. In the cited study, it was shown that ulcers appeared in treatment areas after both calcium electroporation and electrochemotherapy, but none of the ulcers extended beyond the tumor margins, indicating that normal skin was spared less. But not showing anything about the calcium delivery. A similar calcium delivery in normal and cancer cells and tissues has actually been shown in vitro and in vivo.

Methods:

l. 88: it should be clarified that this is added calcium. Or is your 0 mM solution with EGTA or another calcium chelator?

l. 102: Why is there such a wide time span of incubation after electroporation? Does it depend on the experiment? Could it have an effect on the results?

l. 106: delete “to collect the”

l. 126: Why did you not take a fluorescent image before treatment?

l. 133 and l. 136: MTT assay was performed after which experiment?

l. 138: After what specific time point?

l. 140-141: How did you secure that no living cells were removed together with the medium?

l. 144: What control – untreated or 0 mM calcium?

l. 148: After what experiment?

l. 150: What control – untreated or 0 mM calcium?

l. 155-156: How did you test for the assumptions of one-way ANOVA and what did you do in case of violation of the assumptions?

l. 156-157: Why did you transform all percentages?

Results:

Number of individual experiments should be shown in the figure legends.

For each experiment, you should describe in the text and in the figure legends how long after the treatment the experiment was performed, e.g. how long time after electroporation did you measure PI positive cells?

What is your controls? In the Methods (l. 89-90) you write that 0 mM CaCl2 is your control, but at the same time you have “Control” and “0 mM” in figure 1, 5, 9 and 10. In all figure legends you should clearly indicate what your control is – untreated, electroporated in 0 mM calcium, or something else.

It is important to notice that you test small charged molecules, so when you discuss/conclude something about small molecules you should also add that they are charged.

l. 166: “with no calcium” should be changed to “without added calcium”

Figure 2: It is very difficult to see the significance symbols at 800 V/cm in graph A and at 1000 V/cm in graph B.

Figure 3: It is difficult to see some of the significance symbols, e.g. at 18 kV/cm in graph A and B.

l. 220: The number should be changed to mM with one or two decimals.

Figure 4: Indicate in the figure legend what the small windows show. You should also write the reader should note that the axes of the graphs are different (exponential and linear). Use mM calcium in the x-axes since this is the concentration you use throughout the manuscript. You could also add the 0 mM calcium line (the red line) in the large graphs.

Figure 5: What is “successful molecule transfer” defined as? Replace ETBR with EB in the graphs since this is what you use in the text and the figure legend.

Figure 6: Why have you not performed this experiment with nanosecond pulses?

Figure 6: You should be consequent in how you write the y-axis text. When is the fluorescence measured after electroporation? Why is there not high fluorescence when the PI is added at the same time as electroporation “0” (is the axis text correct)? How many single cells did you look at in this experiment (should be added in the legend)? You should also write in the figure legend what the significance stars represent (what lines are compared)?

l. 284: Why did you not use 1800 V/cm since this is what you have use previously and you see a difference in figure 6 using this electroporation parameter?

l. 286: reference is missing

l. 311, 313 and 317: The equations are difficult to read. Sharper versions are needed.

Equation 1: You need to write what each part of the equation is, like you have done for eq. 2 and 3.

l. 324: What is Cm (a number is missing)?

l. 332: “The results of the modelling show…” should be “The results of the modelling (Figure 8) show…”

l. 345-346: Where can this be seen?

Figure 8: Describe what “U” is.

l. 365-366: Please discuss this further. Why the small, less charged molecule?

Figure 9 and Figure 10: These experiments are results and should be moved to the result section.

l. 392: change to “…that was observed in previously published experiments with….”

l. 433-435: There is cell death after electroporation in medium containing 0.25 mM calcium in Figure 10, you show that p<0.01 for this point – what is correct?

Figure 10: When was flow cytometry and MTT performed after electroporation? 0.5 mM calcium is not shown in the figure.

l. 450: Why did you not test the dependence of medium conductivity on cells after nanosecond pulses (similar to the results in figure 9) – now that you see the difference between microsecond and nanosecond pulses?

l. 454: What is the last sentence of the conclusion?

References:

A dash is missing between the page numbers in most of the references.

Please check the volume and page number for reference 20.

Round 2

Reviewer 2 Report

no other comments/suggestions

Author Response

Thank You!

Reviewer 3 Report

The authors have answered my questions satisfactorily and revised the manuscript accordingly, which have improved the manuscript significantly. Thus, I recommend this manuscript being published in Pharmaceutics.

There are only a few very minor comments that I suggest the authors change before publication:

  • L. 20: Delete “calcium” in the beginning of l. 20.
  • L. 22-23: A verb is missing in this sentence.
  • L. 108: Is it 0.51 or 0.50 mM?
  • L. 195-196: Delete one of the “conductivities”
  • L. 678: Change “Ca2+ free medium” to “medium without added Ca2+”

Author Response

The authors have answered my questions satisfactorily and revised the manuscript accordingly, which have improved the manuscript significantly. Thus, I recommend this manuscript being published in Pharmaceutics.

There are only a few very minor comments that I suggest the authors change before publication:

  • 20: Delete “calcium” in the beginning of l. 20.

Thank you for the observation. We have corrected in the manuscript.

  • 22-23: A verb is missing in this sentence.

Corrected in the manuscript.

  • 108: Is it 0.51 or 0.50 mM?

Thank you for the notice. Corrected in the manuscript: 0.5 mM.

  • 195-196: Delete one of the “conductivities”

Corrected in the manuscript.

  • 678: Change “Ca2+ free medium” to “medium without added Ca2+”

Corrected in the manuscript.